# Reduction of Load Capacity of Fiber Cement Board Facade Cladding under the Influence of Fire

**DOI:** 10.3390/ma14071769

**Published:** 2021-04-03

**Authors:** Krzysztof Schabowicz, Paweł Sulik, Łukasz Zawiślak

**Affiliations:** 1Faculty of Civil Engineering, Wrocław University of Science and Technology, Wybrzeże Wyspiańskiego 27, 50-370 Wrocław, Poland; krzysztof.schabowicz@pwr.edu.pl; 2Building Research Institute (Instytut Techniki Budowlanej), Filtrowa 1, 00-611 Warszawa, Poland; p.sulik@itb.pl

**Keywords:** ventilated facades, fire safety, fiber cement board, flexural strength, cladding

## Abstract

The paper analyzes the issue of the reduction of load capacity in fiber cement board during a fire. Fiber cement boards were put under the influence of fire by using a large-scale facade model. Such a model is a reliable source of knowledge about the behavior of facade cladding and the way fire spreads. One technical solution for external walls—a ventilated facade—is gaining popularity and is used more and more often. However, the problem of the destruction during a fire of a range of different materials used in external facade cladding is insufficiently recognized. For this study, the authors used fiber cement boards as the facade cladding. Fiber cement boards are fiber-reinforced composite materials, mainly used for facade cladding, but also used as roof cladding, drywall, drywall ceiling and floorboards. This paper analyzes the effect of fire temperatures on facade cladding using a large-scale facade model. Samples were taken from external facade cladding materials that were mounted on the model at specific locations above the combustion chamber. Subsequently, three-point bending flexural tests were performed and the effects of temperature and the integrals of temperature and time functions on the samples were evaluated. The three-point bending flexural test was chosen because it is a universal method for assessing fiber cement boards, cited in Standard EN 12467. It also allows easy reference to results in other literature.

## 1. Introduction

A ventilated facade is a modern technical solution for the exterior part of a multilayer wall. It consists of an external facade cladding that is mechanically or adhesively attached to a subframe. The subframe is mechanically attached to the exterior structural wall of the building. External facade cladding can be made of a variety of materials, e.g., fiber cement boards, concrete slabs, steel elements, ceramic and other composite elements. Facade cladding is usually installed in accordance with the individual technical design of the facade and the requirements set out by the product manufacturer. They are non-load-bearing elements, bearing only their own weight and environmental impacts such as snow, wind and temperature. External facade cladding does not ensure the airtightness of the building, but only to a certain extent ensures the protection of the external surface of the supporting wall to which the facade is fixed. A ventilated facade is a complete set of individual components that make up a system solution. The standard that sets requirements for a complete ventilated facade system is ETAG 034-1 [1], and the individual components of the whole system must additionally meet national requirements.

The most important element of a ventilated facade is the air gap between the external cladding facade and the insulation layer (mineral wool or stone wool), or the supporting wall if no insulation layer is used. The air gap, also called the ventilation air space, should be at least 20 mm according to ETAG 034-1 [1]; the literature also provides information that the ventilation air space should be in the range of 20 mm to 50 mm [2,3]. It may be reduced locally to 5–10 mm, depending on the cladding and substructure, provided that the performance function of the complete system is not affected. The most important parameter, independent of the dimension of the ventilation air space, is an appropriate possibility of airflow through the air gap. This is ensured not only by the dimension of the ventilation space but also by an appropriate number of ventilation gaps, allowing air to enter this space. Ventilation slots supply air to the ventilation air space, and should be at least 50 cm^2^/1 m facade [1], assuming they are at least at the base point and at the edge of the roof.

Ventilated facades allow the facade cladding to be made with different materials, structures, textures, or colors. Due to good aesthetics and durability, a ventilated facade is increasingly used as a technical solution for the external multilayer walls of newly constructed buildings, but it also performs well in the case of buildings undergoing renovation. External facade claddings can be made of very large elements, e.g., the standard size for fiber cement boards is 1.25 × 3.10 mm^2^ [2], and for HPL (high-pressure laminate) boards, 1.85 × 4.10 mm^2^ [2].

This paper analyzes a ventilated facade with external facade cladding made of fiber cement boards, which are classified as fiber-reinforced composites. These composites are characterized by two phases [4,5]. The first phase is a cement matrix, based on Portland cement. The second phase of these composites is the dispersed phase, which is characterized by the distribution of fibers in a discontinuous and randomly oriented manner. In the case of fiber-reinforced composites, they offer many of the benefits of using fibers under normal conditions [6,7]. In fiber cement board mainly cellulose fibers, polyvinyl alcohol (PVA) synthetic fibers and polypropylene (PP) fibers are used. Unfortunately, in the case of fire conditions, there are few such studies, which is due to the complexity of the tests and their costly nature. The individual fibers have the following melting points: PVA (polyvinyl alcohol) synthetic fibers, about 200–220 °C [8,9]; PP (polypropylene) fibers, about 160–175 °C [8,10]; cellulose fibers, 260–270 °C [11].

Fiber cement boards have not been extensively studied in terms of fire temperatures and their behavior on the facade in case of fire. Szymkow’s research [12] showed that the fibers in fiber cement boards are destroyed at 230 °C after about 3 h of exposure. In ref. [13], a decrease of about 10% in the compressive strength of concrete and fiber concrete was seen at temperatures up to 300 °C. In contrast, for fiber-reinforced cement composites, the flexural strength increases with increasing temperatures, up to about 300 °C [14]. In the case of compressive strength in fiber concrete and fiber cement, this temperature rise does not reduce the compressive strength; on the contrary, it increases the strength by evaporating water from the pores (this is confirmed by tests [8] in which high temperatures acted upon the fiber concrete sample for about 100 min). Temperatures in the range of about 300 °C, as shown in the above tests, are dangerous only for the fibers because their melting point is exceeded. Temperatures higher than 400 °C, as presented in the research by Szymkow [12], only strengthen the cement matrix over a short period of time (usually in the range of 2.5–7.5 min, depending on the sample). During this time, water evaporates from the pores, which increases the bending capacity, but after this period the fibers begin to melt and then the strength drops drastically.

Unfortunately, the temperatures of a fire affecting external facade claddings may locally reach values even exceeding 800 °C. The external curve reflecting a developed facade fire, as presented in the standard [15], represents values that limit the temperature to about 660 °C. Nevertheless, higher temperatures may be expected in the vicinity of the window lintel. Unfortunately, experimental studies are not available for fiber cement boards. Looking at the above analogies of other fiber materials, interesting conclusions are reached in ref. [13], where tests were performed for concrete and fiber-reinforced concrete. The compressive strength of concrete and fiber-reinforced concrete of class C30/37 in temperatures of 800 °C reduces by more than 90%. At 500 °C and 600 °C, the samples without fiber addition were destroyed during their annealing, while those with polypropylene fiber additions retained residual flexural strength [14]. In ref. [16] the positive effect of using fibers to increase the flexural strength of beams subjected to a normative fire temperature curve is demonstrated. Research shows that, despite the fact that the fibers are subject to destruction at fire temperatures, the voids thus formed allow the material to withstand higher temperatures afterward. The emergency situation of a fire is shown in Figure 1.

Ventilated facades, compared to ETICS (External Thermal Insulation Composite System) facade, in terms of the problems of falling facade elements during a fire, show much worse parameters [17,18,19]. The emergency units, which are responsible for evacuating people inside the building in the event of a fire, are particularly at risk. This problem is well known in the scientific community, and the work of the European Commission is also based on solving this problem [20,21].

There are several methods for testing the response of fiber cement boards to high temperatures. One of them was presented by Szymkow in ref. [12]—by annealing the samples in a furnace, specially prepared for this test. A similar way of affecting fiber cement board samples was adopted in ref. [22]. It is also possible to prepare a large-scale facade model. As far as this form of testing is concerned, there are many standards in the world for testing such models [20,23,24,25]. In most cases, they assume a fire spreads from an opening towards the facade, and simulate the window openings of a room. A hearth (fire source), defined by a temperature action standard curve, is located in a recess. Flames escape from the opening, affecting the facade and other wall elements. The standards differ in details, i.e., the type of hearth (wood cribs [20,23,24] or gas [25]), the opening dimensions, the test time, and the large-scale facade model and its dimensions. A comparison of different standards for testing life-sized facade models for fire safety is summarized in ref. [26].

## 2. Materials and Methods of Ventilated Facades

By analyzing the literature, deficiencies were noted in the response and destruction of fiber cement boards when exposed to high temperatures through fire. The authors decided to verify how the high temperatures from the action of fire affect the reduction of flexural strength of fiber cement board on the facade.

To perform the study, the authors prepared a model of the facade. For this purpose, they used a large-scale facade model. The facade to be analyzed was attached to a test platform made of autoclaved cellular concrete blocks, traditional masonry and poured reinforced concrete lintels.

The facade was made of 8 mm-thick fiber cement boards of natural color—not pigmented. The cladding was fastened using mechanical connections to a galvanized steel substructure. The substructure was mechanically attached to the platform through consoles. The division of the facade into facade cladding is shown in Figure 2.

The full view of the large-scale facade model, as it was conducted, is shown in Figure 3a. In addition, the sand burner and the combustion chamber can be seen, through which the fire scenario is implemented. The facade cladding in the left and right part is fixed in a different way, which causes the left part to protrude beyond the face of the plane—this can be seen in Figure 3b. The technical solution for filling and maintaining the stone wool and ventilation space is shown in Figure 3c.

The impact of high temperatures on the facade was tested by means of fire and fire gases escaping from the combustion chamber—the scenario of fire in the room and fire escaping through the window opening onto the facade was carried out. The combustion chamber on the back wall contained a blower, which allowed us to reproduce the real conditions of a fire. The fire source was a sand burner that used gas as fuel—appropriately metered through a mass flow meter. The sand burner used liquid propane of 95% purity (PGNiG Polska) as fuel, with a release rate scaled in such a way that it corresponded to the fire development in accordance with the external curve, which is characteristic for facade fires as cited in EN 13501-2 [27]. International fire curves show the temperature course over time at the source location. The international fire curves, including the external curve, are shown in Figure 4. Laminar air inflow from the side of the room was provided by honeycomb.

The large-scale facade model was equipped with a set of thermocouples (named TE1-TE9) to verify and identify the damage to the exterior cladding. They were placed in accordance with the expected development and impact of flames and hot fire gases. Thermocouples were placed on the surface of the board cladding—in 3 rows at heights of 800 mm, 1600 mm and 2500 mm above the combustion chamber. These places were selected due to the expected shape of the flame coming out of the combustion chamber. The exact arrangement of thermocouples and their names are shown in Figure 5.

## 3. The Test and Results

The test scenario of heating with the sand burner corresponded to the intensity of a fire equivalent to that which would be experienced if a real fire occurred in the room located directly behind the facade (inside the building), with flames escaping from the window and affecting the facade.

The test was conducted at an ambient temperature of 22.3 °C and a relative humidity of 52%. The test began by setting up the burner and properly calibrating the supplied gas. The beginning of the test is shown in Figure 6a—uniform distribution of flames leaning against the splay is noticeable. Such a view is characteristic of the beginning of the test, where the model is not exposed to external winds and the facade cladding is not yet degraded in the initial phase. The first 6-min period of the fire brought smoke/charring of the external facade cladding, and cracks appeared on the upper splays, which are most exposed to high temperatures. Facade cladding mounted on the splays fell off first, after a time range of 9 min to 14 min. During this period, the destruction of the external facade cladding progressed, as shown in Figure 6b, where the continuing degree of destruction is shown on the already targeted facade cladding, within 600 mm of the bottom. In Figure 6c,d, the progression of facade cladding deterioration under the effect of fire temperatures can be seen; in the minutes that followed, the external facade cladding elements fell off. The side splay boards fell off at about minute 17.

At the end of the test, the load-bearing capacity of the fiber cement boards on the higher elements was progressively exhausted as more and more elements fell off—this is shown in Figure 7a,b. Figure 7c shows the appearance of the cladding after the 60-min test. Significant deterioration of the cladding on the right side of the test model is evident.

As shown in Figure 5, the thermocouples recorded the temperature continuously during the test. The results of these tests are shown in Figure 8. It can be observed that the temperatures for TE1 and TE5 thermocouples, and TE6 and TE8 thermocouples, are similar, operating in the low-temperature range—below 100 °C. Such temperatures have no destructive effect on fibers in fiber cement boards. The whole material did not show major signs of wear at these temperatures, either. In the case of TE2, TE3, and TE4 thermocouples a disproportion is visible, i.e., much higher temperatures prevail on the right side of the large-scale facade model—this is due to a smaller protrusion of the cladding beyond the face. Although this difference is minimal (20 mm), it directs all the flames to the right side where the TE4 thermocouple is located. It is also noticeable that the flame source has a much greater effect on thermocouples located in a non-central position, such as TE2 and TE4, than on thermocouples located centrally but higher up, such as TE7.

## 4. Materials and Methods

In order to investigate the scientific issue, which was to verify the reduction in flexural strength of the fiber cement boards during fire impact, samples were taken from the model. Samples were taken directly from the large-scale facade model or from cladding elements that fell off. Samples were taken from 3 locations, shown in Figure 9, and compared to reference samples.

The method in which the samples were taken and how long they were exposed to the fire is presented below:Sample D5—fell off in approx. 13’30” of the test;Sample D4—fell off in approx. 17’15” of the test;Sample D3—fell off in approx. 34’00” of the test.

Modulus of rupture (*MOR*) assessment was performed according to PN-EN 12467 [28]. The dimensions of the reference samples were taken as 250 × 250 mm^2^. For the other tested samples, the dimensions recommended by the standard [28] could not be achieved due to the recovery of the samples from the elevation model after the test and the extensive damage to these elements. The three-point bending flexural test stand is shown in Figure 10.

The results of *MOR* for the reference samples are shown in Figure 11—the significant difference between the courses of the individual graphs for different samples is evident. The direction in which the samples were bent in the test (whether they were cut parallel or perpendicular to the pressing direction) was important for the test results. In the case of bending in the direction perpendicular to the pressing direction of the boards (the samples “001-D.ref” and “002-D.ref” show a much higher flexural strength), the destructive force is higher. In the case of bending in the direction parallel to the pressing direction of the boards, samples “003-D.ref” and “004-D.ref” have lower flexural strength.

A total of 5 samples were taken from the large-scale facade model, and all the results of the strength test course are shown in Figure 12. All the samples are characterized by a significant reduction in strength under the three-point flexural test. In addition, they are characterized by a very rapid and sharp reduction in flexural strength after passing through the point representing the destructive forces.

The flexural strength was calculated—*MOR* (f_max_)—according to the formula in the standard [28]:(1)$$MOR\;=\;\frac{3Fl_{s}\;}{2b\;e^{2}}$$
where:*MOR*—modulus of rupture (MPa);*F*—the load (force) (N);*l_s_*—the length of the support span (mm);*b*—sample width (mm);*e*—sample thickness (mm).

The modulus of elasticity was determined based on the load of proportionality (LOP), which is the limit of applicability of Hooke’s law, determined using the graphical method presented in ref. [22]. The bending modulus of elasticity was determined from the Equation:(2)$$E_{D}\;=\;\frac{Fl_{s}^{3}}{4fbe^{3}}$$
where:*E_D_*—Young’s modulus (GPa);*F*—the load (force) (N);*l_s_*—the length of the support span (mm);*f*—flexion (mm);*b*—sample width (mm);*e*—sample thickness (mm).

Table 1 shows the previously calculated strength parameters, and their mean values were calculated.

## 5. Discussion

D4 samples taken from a height of approximately 700 mm above the top splay above the combustion chamber show better flexural strength than the previous sample. The destruction occurs very rapidly in the testing machine, as shown in Figure 12. Unfortunately, as with the above sample, these materials do not resemble the non-degraded material. For the two samples taken, there is a significant difference between their strengths, being 10.26 MPa and 3.86 MPa, respectively. For the elasticity moduli, these values are similar. These samples were taken from a height of about 700 mm above the top splay, which roughly corresponds to the TE3 thermocouple and the temperatures shown on it (see Figure 7).

Temperatures as high as 600 °C result in the complete destruction of the sample. Cladding components in this area fall off after about 17’15”, which corresponds to an integral of the time-temperature function of about 3.8 × 10^5^ (s·°C). The moment at which the D4 samples fall off along with the integral is shown in Figure 13.

D3 samples are from an area approximately 1200 mm above the top splay above the combustion chamber. Samples were also taken from elements that fell off the large-scale facade model. These persisted for approximately 34’ until they fell off. D3 samples show greater strength stability than those taken at locations D4 and D5. The results are shown graphically in Figure 14—despite the increase in modulus of elasticity and *MOR*, they still show reduced load capacities compared to the reference samples. In order to evaluate which integral of temperature and time function acted on it, the integral of function and time was interpolated between TE3 and TE7 thermocouples—it corresponds to the value of approx. 6.7 × 10^5^ (s·°C). The results are presented in Table 2.

Temperature graphs for TE3, TE7 and TE9 thermocouples with their corresponding time and temperature function integrals (s·°C) are shown in Figure 13; the timeline also shows how long it takes for the individual facade elements sampled to fall off.

The results presented in Table 1 were carefully analyzed. As shown in Figure 14, samples subjected to fire, even 1200 mm above the upper splay, do not show sufficient strength, and their modulus of elasticity shows reduced values. A tendency can be noticed that the integral of the temperature and time functions, despite a significantly greater impact on D3 samples, does not show a lower elasticity modulus, and only the flexural strength is reduced. It can therefore be assumed that with such a large impact on this function, the samples remain on the facade only because of their internal predispositions, e.g., the proximity of mechanical fasteners (rivets). The samples taken at a height of approx. 200 mm or approx. 1200 mm above the splay show low flexural strength parameters.

## 6. Conclusions

The large-scale facade model is a great source of knowledge regarding the behavior of facade cladding during a fire and the way the fire spreads. The problem of facade cladding destruction during a fire in terms of different materials is insufficiently recognized. The authors used fiber cement boards, i.e., fiber-reinforced composite materials, as facade cladding.

In the analysis of the reduction in the load capacity of fiber cement boards, two key elements must be distinguished: the time for the element to fall off the facade cladding, and the degree of degradation. In the case of fire-induced detachment of degraded elements, temperature and exposure time are crucial, with the result that lower portions fall off first—even though the fiber melting point is exceeded over a larger area of the facade. In addition to temperature, the time at which the elements fall off is also influenced by the extent to which the cladding protrudes beyond the face of the entire facade. There is a noticeable tendency for the fire to spread in the direction where the cladding is closer to the supporting wall structure. The first few minutes consist of charring and destruction of the cladding in the model. The first pieces of significant size fall off the facade at minute 17.

The elements that fell off and that were taken for testing show insufficient flexural strength for their further use. It is also noticeable that the fragments of facade cladding that fall off from higher parts are less degraded, even though the integral responsible for temperature and time function is significantly larger. A higher temperature (in the fire range) has a more destructive effect than a temperature of about 200 °C or lower, over a much longer time period, where the integral of the time–temperature function is about 44% greater.

In terms of flexural strength, the effects of higher fire temperatures are also more significant than duration. Samples from lower parts of the model had lower values of modulus of elasticity and flexural strength.

All samples taken up to a height of about 1300 mm above the level of the combustion chamber are not suitable for reuse. Their further use in such a state would pose a significant risk to people moving beneath such an elevation.

The tests carried out showed the distribution of reduced flexural strength of fiber cement board cladding, fixed in different parts of the tested model, under the influence of fire.

The authors plan further research using large-scale facade models. Potential possible studies include an analysis of the time it takes for the fibers to degrade inside the cement matrix, and further, more global research on the behavior and depletion of load capacity.

## Figures and Tables

**Figure 1 materials-14-01769-f001:**
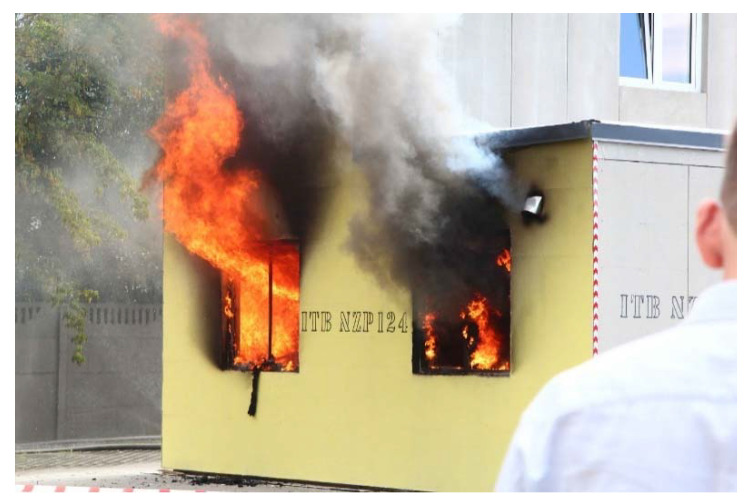
View of flames escaping from a room involved in a fully developed standard fire, during a field test. (Author: E. Kotwica).

**Figure 2 materials-14-01769-f002:**
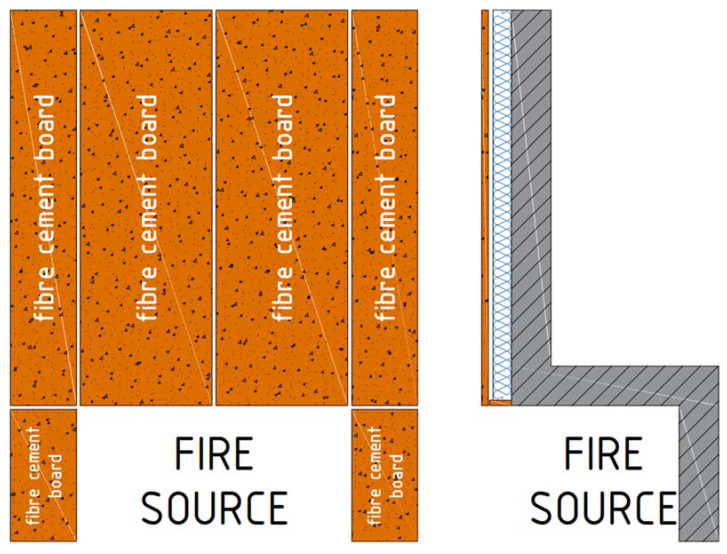
Division of boards on the facade—front and side view.

**Figure 3 materials-14-01769-f003:**
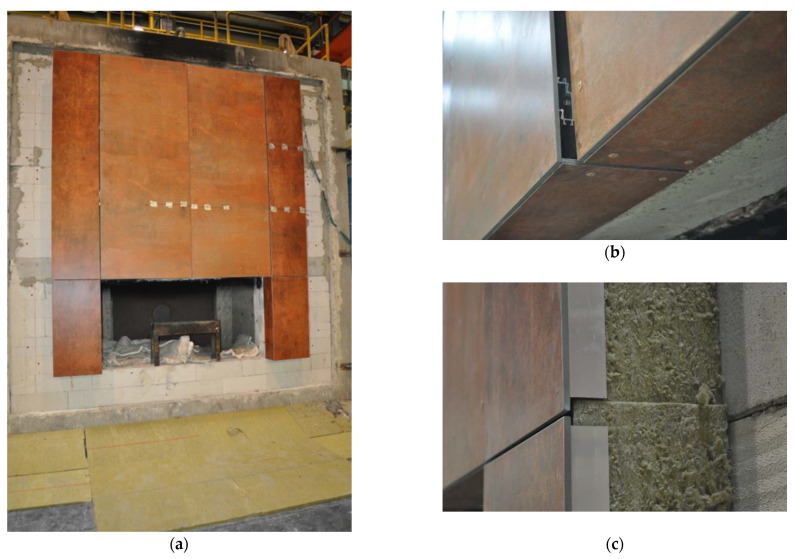
Selected details of the large-scale facade model: (**a**) actual elevation model view; (**b**) eaves problem and solution; (**c**) insulation.

**Figure 4 materials-14-01769-f004:**
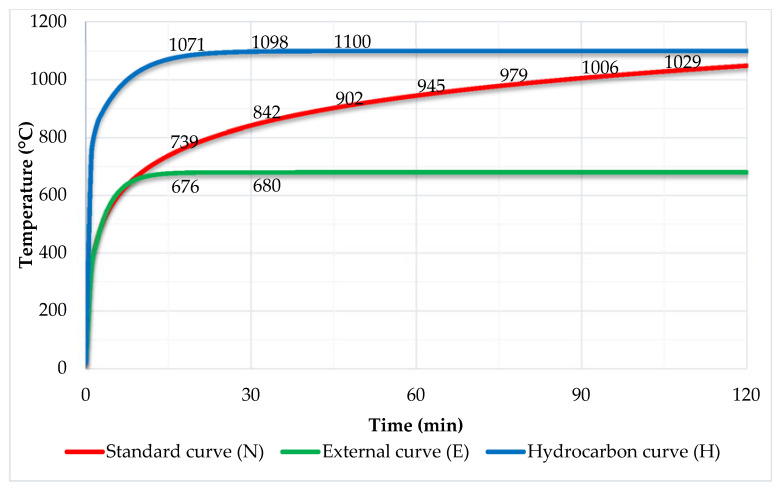
Summary of fire dependencies.

**Figure 5 materials-14-01769-f005:**
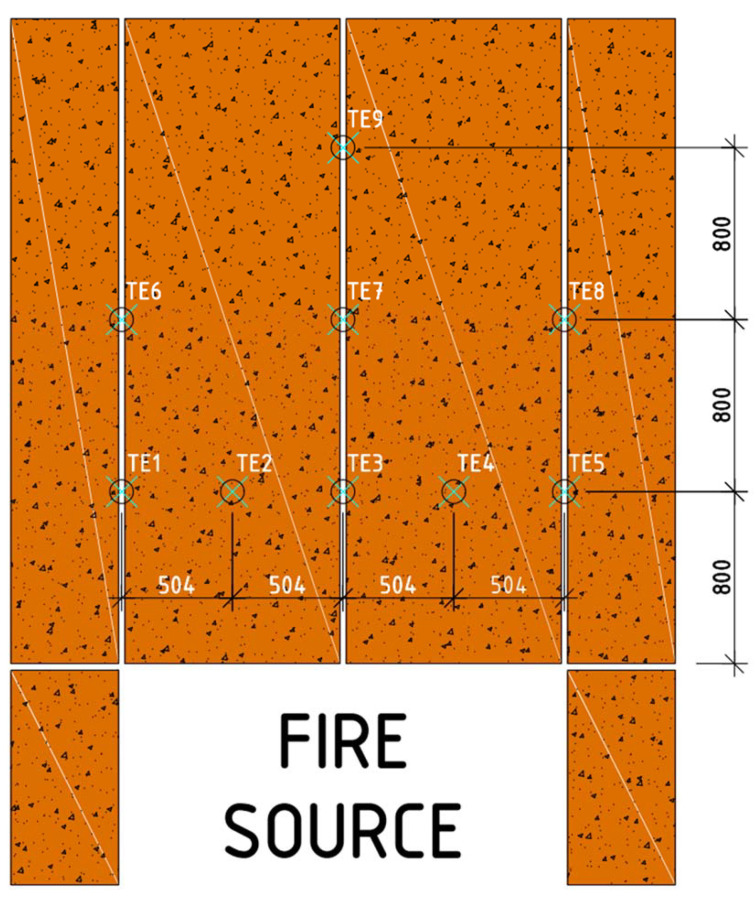
Location of thermocouples on the large-scale facade model—front view.

**Figure 6 materials-14-01769-f006:**
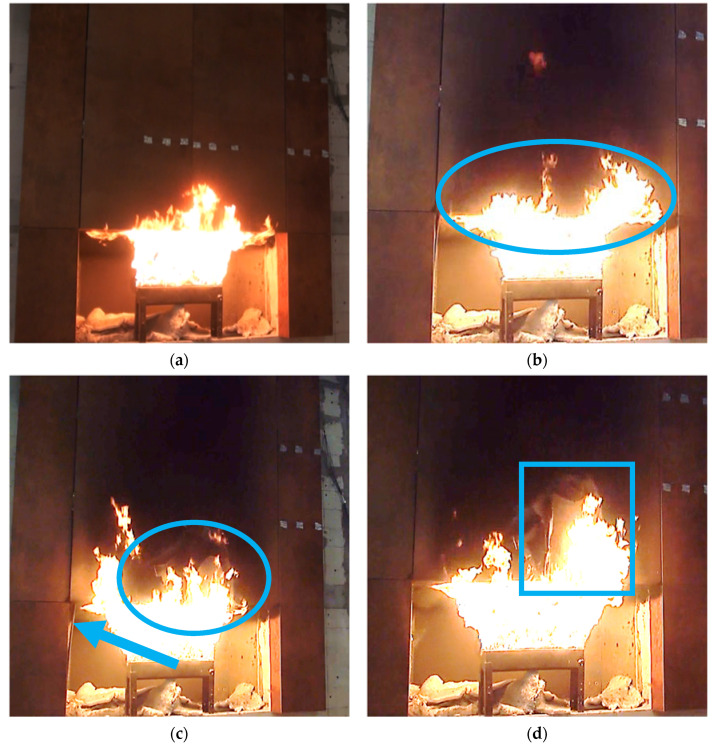
View of the large-scale facade model in successive minutes of the test: (**a**) start of the test; (**b**) approximately 11’00”—visible degradations; (**c**) approximately 17’00”—splay piece falling off; (**d**) approximately 20’30”—progress of destruction.

**Figure 7 materials-14-01769-f007:**
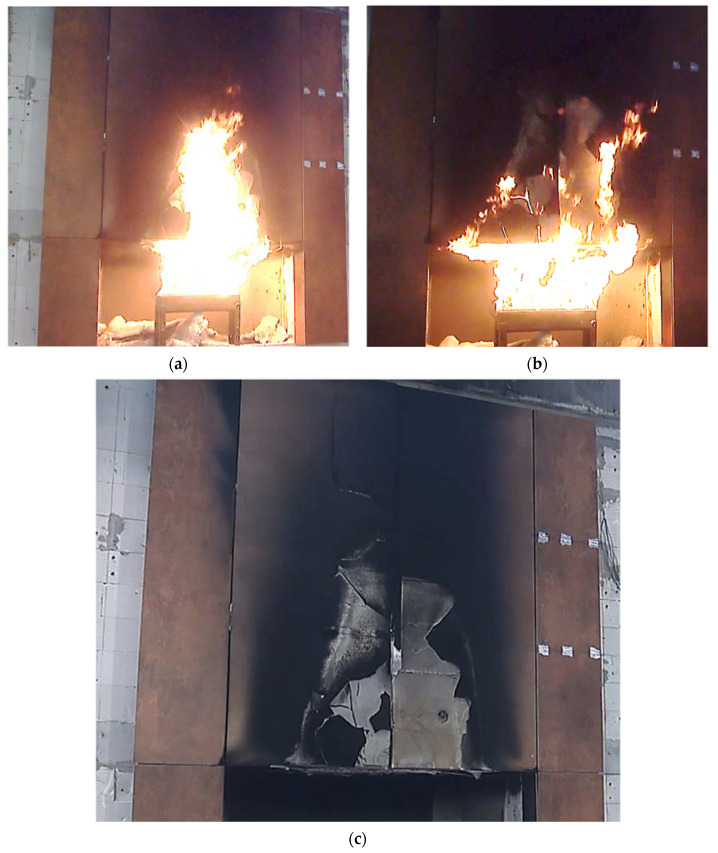
View of the large-scale facade model in the subsequent minutes of the test: (**a**) approximately 40’00”—high cladding degradation; (**b**) approximately 55’00”—more elements are falling off; (**c**) view of facade cladding after testing.

**Figure 8 materials-14-01769-f008:**
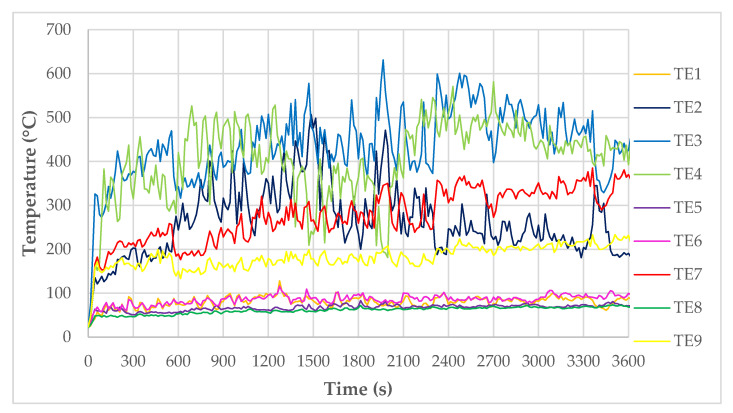
Temperature measurement results for thermocouples on the large-scale facade model.

**Figure 9 materials-14-01769-f009:**
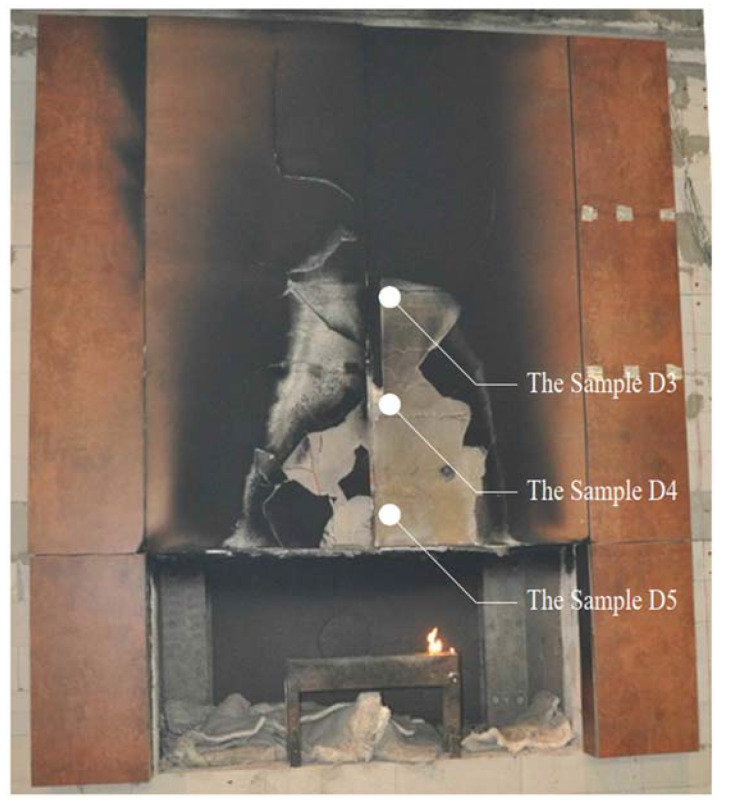
Sampling location.

**Figure 10 materials-14-01769-f010:**
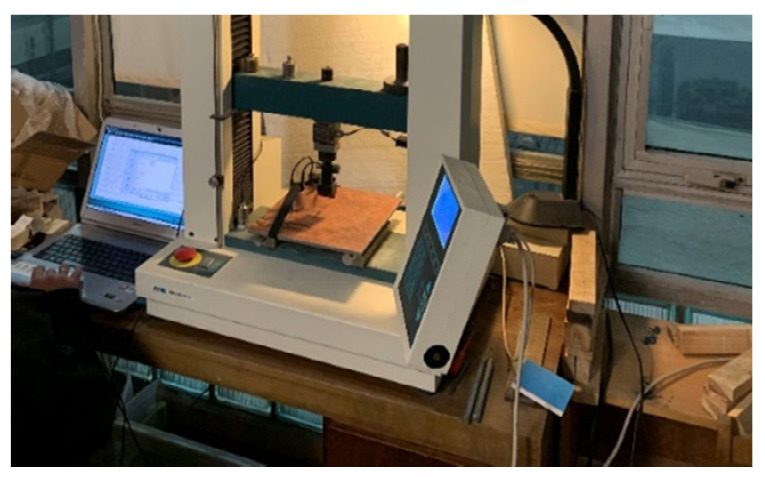
Three-point bending flexural testing machine.

**Figure 11 materials-14-01769-f011:**
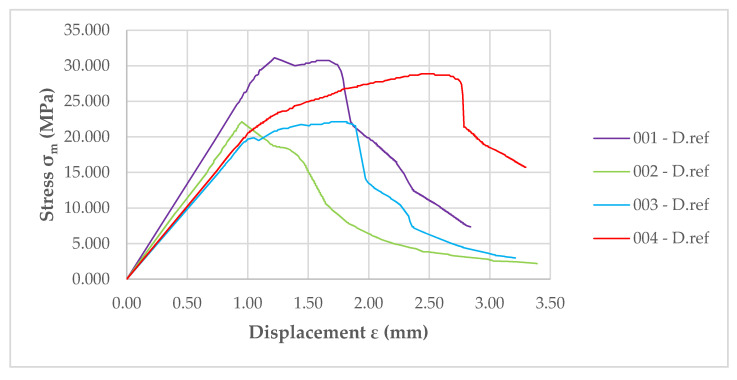
Graph showing the modulus of rupture (*MOR*) of the reference samples.

**Figure 12 materials-14-01769-f012:**
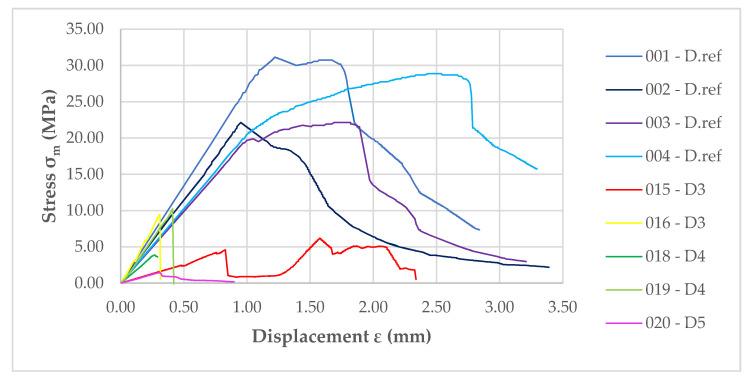
Graph showing *MOR* of front facade samples compared to reference samples.

**Figure 13 materials-14-01769-f013:**
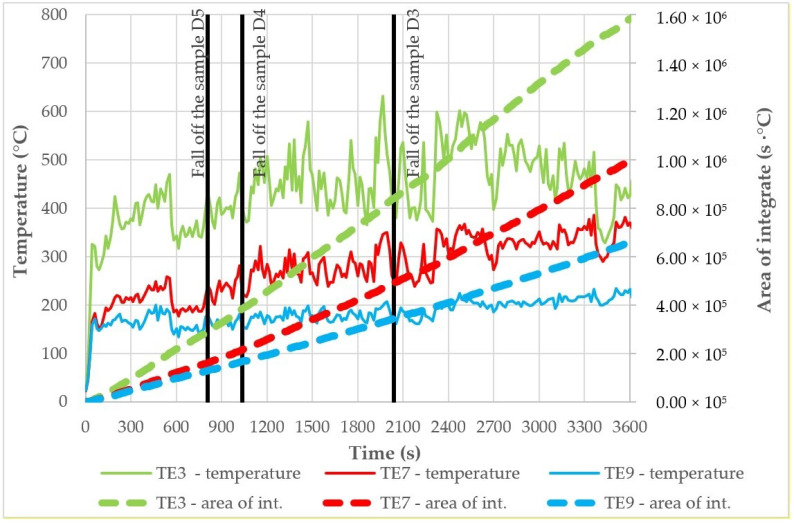
Temperature measurement results for TE3, TE7, and TE9 thermocouples, and the increasing integral for the temperature–time function, along with the determination of the falling-off time of individual elements.

**Figure 14 materials-14-01769-f014:**
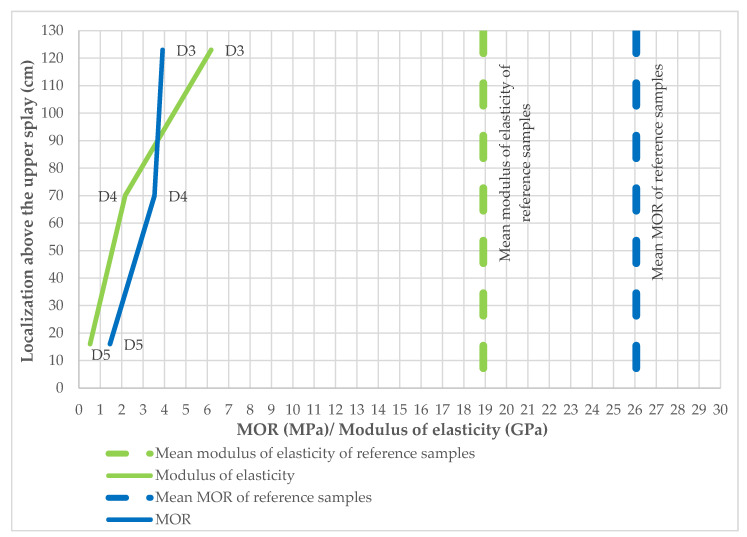
Reduction in MOR of the fiber cement boards depending on the localization above the upper splay.

**Table 1 materials-14-01769-t001:** *MOR* and modulus of elasticity for individual samples.

Sample Identification	*MOR* (MPa)	Modulus of Elasticity (GPa)	Mean Value *MOR* (MPa)	Mean Value Modulus (GPa)
D.ref.	31.13	22.49	26.06	18.91
D.ref.	22.13	19.39		
D.ref.	22.13	16.61		
D.ref.	28.88	17.17		
D5	1.45	0.52	1.45	0.52
D4	10.26	2.70	3.53	2.16
D4	3.86	1.61		
D3	9,45	7.88	3.91	6.18
D3	6,20	4.48 *		

* Secondary the flexural strength when the sample is supported at all edges.

**Table 2 materials-14-01769-t002:** Characteristic data of falling samples.

Sample Identification	Time of Fall of the Sample (s)	Area of Integrate (s·°C)	Temperature’ Thermocouples in Time of Fall of the Sample (°C)
D5	810	>2.9 × 10^5^	>437
D4	1035	3.8 × 10^5^	403
D4			
D3	2040	6.7 × 10^5^	328
D3			

## Data Availability

The data presented in this study are available on request from the corresponding author.

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
