# Peer review of "Reduction of Load Capacity of Fiber Cement Board Facade Cladding under the Influence of Fire"

_materials, 2021, doi:10.3390/ma14071769_

Round 1
Reviewer 1 Report
The paper presents the results from a spectacular experiment using flammable materials with the purpose to demonstrate that high temperatures reduces the mechanical properties of fiber-cement board facade cladding. Although the results are in line with the expectations, hence no to little novelty could be encountered in this regard, the model and the data presented in the paper might be useful in some particular situations.
Some other observations are presented further:
- line 56: please define the acronym HPL before using it;
- lines 113-115: this last phrase is redundant to the phrase from lines 88-89 and might be excluded;
- lines 124-125: facade or façade?;
- line 134: please rephrase;
- figure 4 deserves a better explanation;
- some details (technical data of the model and the procedure used during testing) about the stand used in three-point bending flexural test could be presented in Materials and methods chapter;
- some hints regarding future work needed on this topic could be included in the Conclusion chapter.
Reviewer 2 Report
The manuscript presented investigation works and findings about fire testing of fibre-cement board facade cladding and the reduction in flexural strength. The results are interesting and are of good reference value to the readers. The quality of the manuscript is satisfactory and several corrections are necessary for its possible acceptance. The authors should explain the merits, rationale, and impact on fire safety of using ventilated facade in buildings. A table should be included to summarise the experimental samples identification. The format of citations and references should be rectified, and English translation of Polish titles should be included, for example in page 3 of the manuscript and the reference list. In Figure 5, the level of the thermocouple at top does not match with the description in the text. In Figures 11 to 13, the legends of graphs should be improved. For the mathematical symbols, the subscripts of symbols should be properly displayed.
Reviewer 3 Report
The manuscript entitled “Reduction of flexural strength of fibre-cement board facade 2 cladding under the influence of fire” reported as an original article on cement board façade under fire resistance. The paper is very interesting piece of research but required many changes and modifications.
Title
The title could be improved to better wording, I will leave this to authors how he could improve it. It also no need to have a dot at the end of the title. Type of fiber is great to be mentioned here.
Abstract
Line 12: it is very confusing sentence, required to reword. Or divide the sentence into two parts until deliver message clearly.
Line 20-21: it is not clear why authors choose only flexural strength test for fire testing, it should give a reason.
Overall, the abstract can be written in better way to attract reader to read the paper. I could not see much information about importance of the cement-board.
Keywords:
Please add few more keywords, like flexural strength, cladding
Introduction
Line 28: by cladding fixing, it sounds wrong and strange. Please revise.
Line 30: authors mean fiber reinforced cement board, such as which type of fibers (glass fiber, carbon fiber ) so on. Please mention type of fibers there are many fibers.
Line 33: authors should give a plenty of background and earlier studies that have been done on fiber reinforced cladding such as this paper (Glass Fibre Reinforced Concrete Use in Construction). Please give earlier face cladding before go into details.
Line 41; authors mentioned “ insulation layer” what is the insulation layer, authors should give a description before just entered into the paper.
Line 49-50: authors should give some details on the gap, how is look like and how many recommended by 1 SQM or Cubic meter. It could be their own words.
Line 55-56: that is not shocking, everythings nowadays are prefabricated, so what authors try to say it is not clear, did authors try to easy handling, easy transporting, easy application and installing, very strong in bending and flexing. Please explain.
Line 62: please supported with evidence of research and publication. I have seen mostly with GFRC. These fibers are given by the authors, their melting points are so low. If authors check for glass fiber has much higher melting point and more usuable in cladding than any others fiber, please refer to paper (Glass Fibre Reinforced Concrete Use in Construction) for more information, which has been used and applied in many places in India.
Line 68-69: why the destruction is happened only in the flexural testing? Is it a good representative for the exposure of the fire? This is need further details.
Line 70-71: required to reword, it is not clear enough, so it means at the elvated temperature of 300oC the compressive strength would loss their 10% of their strength.
Line 72: is that really true? I have a doubt, unless the fiber have a greater melting point than 300oC.
Line 74: authors when it mentioned safe for the short term, should mention the duration whether 10 minute an hour. Please be precise.
Line 80: here again, please give the numbers instead of guessing time. Please revise.
Line 90-100: authors should revise and write in a clear way, so many confusing words. Please try to chop and reduce the length of the sentence.
Figure 1: it is better to come after sentence to high temperatures, such as … (line 99). Authors can show a better image without shoulder of human in the photo.
Line 106-115: I was expecting to authors gives extra information on the fire how it works in the cement-board matix while reinforced with fiber (please mention type of fiber) and at melting evaporated the plastic materials make a porous in the materials matrix before spalling or explode. Please revise.
Figure 2: it is not clear, how they put the fiber cement board, please show in 3D or give the right angle of explantion. Authors might mention top view or side view because it is not clear, I saw the panel on the floor instead of standing surrounding the fire source. Please fix.
Figure 3 c. what is the type of insulation ? authors earlier just talked about air not such insulation. Please explain.
Figure 4: this figure is given, but there is not any explation about the figure, for example what is external cuve(E), and other curves. Please explain or remove.
Figure 5: please explain the side of view, I feel it is side view, according to the figure 3, however, authors should mention that. I have also wonder if the fire happens usually will be around the fire, these panel are located in the top of the hearth. Explain.
Figure 6 and 7: it is ok photos, but authors could make them into a chart shows the effectof temperature with time and spalling of the cement board until spalling off.
Line 193: authors mentioned the the figure 5 without giving an explaition. Please fix and write in the figure 5 shows the position of the thermocouples. I have just wondered why these location have been selected? Please revise.
Figure 5: in the figure 5 I could not see the TE 2, 3…8 Please revise, it is all written 1.
Line 222: why the 250x250 mm? is that for compressive strength or flexural ? according to the topic, it should be flexural. This dimension is for compressive. And please mention which standard are followed.
Line 240: why authors used three point bending test not four point bending ?
Line 241: fix the “influxural”
Figure 12: why the point D5 have only one result? It suppose all to be three.
Conclusion: please specify all in bullet points, revise all. There is also some plagiarism detection please fix this issue in the paper.

Round 2
Reviewer 3 Report
In this format, it is ok. I feel the conclusion, can be presented in a better way rather than all detail just give a bullet point of highlighted outcomes.